## RESEARCH ARTICLE

# Phenotypic plasticity shapes carry-over effects in sea rock-pool mosquitoes

Giulia Cordeschi[1,2,*], Roberta Bisconti[2], Valentina Mastrantonio[1], Daniele Canestrelli[2] and Daniele Porretta[1]

## ABSTRACT

Environmental conditions during early life can shape trait expression after metamorphosis. Direct carry-over effects occur when the value of a trait expressed at an early stage directly determines its expression at later stages, maintaining phenotypic continuity across metamorphosis. However, environmental factors affect multiple traits whose interaction might shape developmental trajectories. Here, we tested whether trait interactions can shape direct carry-over effects, examining the interplay between behavioural and morphological plasticity in the mosquito *Aedes mariae* under varying salinity conditions. We found that higher salinity caused a reduction in larval body size and an increase in resting behaviour at the water surface, at the expense of browsing activity. Furthermore, we found that larval body size was positively correlated with pupal size under constant conditions, indicating a direct carry-over effect. However, this relationship was disrupted as salinity increased, due to different behavioural response according to larval body size, which decouples pupal morphology from larval size. Our results show that environmental conditions modulate trait integration and modify direct carry-over effects. These findings highlight the importance of considering multiple traits when studying developmental plasticity and contribute to the debate on the extent to which one life stage is coupled to the others across the metamorphic boundary.

KEY WORDS: Direct carry-over, Multivariate plasticity, Behavioural plasticity, Morphological plasticity, Complex life-cycles, Water salinity, *Aedes mariae*

## INTRODUCTION

The environment varies across space and time, and phenotypic plasticity is one of the major mechanisms enabling organisms to respond to such variability (Fox et al., 2019; Scheiner et al., 2020; Yeh and Price, 2004). Phenotypic plasticity is the ability of a given genotype to respond to environmental change by expressing different phenotypes (Pigliucci et al., 2006; West-Eberhard, 2003) and it can affect virtually any trait including morphology, physiology, behaviour, and life history (Whitman and Ananthakrishnan, 2009). Some emblematic examples of phenotypic plasticity include castes in social insects (Miura, 2005), polyphenism in migratory locusts

(Applebaum and Heifetz, 1999), seasonal polyphenism in caterpillars and butterflies (Brakefield and Frankino, 2006), alteration of generations in aphids (Brisson, 2010), and alternative strategies for reproduction in beetles (Moczek, 2010), among others. A particular form of developmental plasticity is a phenomenon known as carry-over effect, where the environment an organism experiences early in life can impact its traits and performance at a later time (Harrison et al., 2011; Marshall and Morgan, 2011; Moore and Martin, 2019; O'Connor et al., 2014; Pechenik, 2006). These effects can arise in a variety of contexts – including within and across life-history stages, between seasons, or across experimental conditions (O'Connor et al., 2014). In animals undergoing metamorphosis, juveniles and adults are separated and although metamorphosis is thought to decouple these stages from one another (Aguirre et al., 2014; Moran, 1994), environmental conditions experienced early in life can shape trait expression across this event of major change (Collet and Fellous, 2019; Hufford et al., 1999; Kasumovic, 2013; Moore and Martin, 2019; O'Connor et al., 2014; Pechenik, 2006; West-Eberhard, 2003).

When environmental factors influence the expression of a trait in an earlier life stage and this effect is directly transferred to the same trait in subsequent stages, it is referred to as a 'direct carry-over effect' (Moore and Martin, 2019). Typical cases of direct carry-over effects are size at metamorphosis in marine invertebrates, insects and amphibians, where larval body size determines post-metamorphic body size (Allen and Marshall, 2013; Boes and Benard, 2013; Moore et al., 2018). For example, in wood frogs (*Rana sylvatica*), canopy cover affects larval size, with individuals from closed-canopy ponds growing larger than those from open-canopy ponds – an effect that persists after metamorphosis (Boes and Benard, 2013; Halverson et al., 2003).

Nevertheless, more complex patterns have recently emerged (Arambourou et al., 2017; Bouchard et al., 2016; De Block and Stoks, 2005; Debecker et al., 2015; Zamora-Camacho et al., 2022). Indeed, organisms often respond to environmental change with plasticity in multiple traits (i.e. multivariate plasticity) (Foster et al., 2015; Schlichting, 1989). Well-documented examples include predator-induced defences in aquatic animals, where prey modify behavioural, morphological, and life-history traits in response to predator presence (Kishida et al., 2010; Relyea, 2004; Spitze and Sadler, 1996). Moreover, plastic responses in one trait can, in turn, shape the development and selection of other traits (reviewed in Nielsen and Papaj, 2022). For instance, the caterpillar *Battus philenor* responds to high temperatures by changing its body colour from black to red and leaving host plants to seek cooler locations for thermal refuge (Nice and Fordyce, 2006). Experimental studies have shown that red colouration reduces the frequency of refuge-seeking, suggesting that the interaction between these two plastic traits is mediated by the effect of colour on body temperature (Nielsen et al., 2018). This multivariate perspective has provided valuable insights into the trade-offs between the costs and benefits

[1]Department of Environmental Biology, Sapienza University of Rome, 00100 Rome, Italy. [2]Department of Ecology and Biology, Tuscia University, 01100 Viterbo, Italy.

*Author for correspondence (giulia.cordeschi@uniroma1.it)

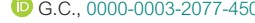 G.C., 0000-0003-2077-4508

of plasticity, shedding light on evolutionary constraints, adaptation potential, and how populations persist in rapidly changing or novel environments (Ellers and Liefting, 2015; Lande, 2009; Matesanz et al., 2021; Morel-Journel et al., 2020; Nielsen and Papaj, 2022). However, when considering plasticity across metamorphosis, the extent to which interactions between multiple plastic traits in early life stages influence the magnitude of carry-over effects remains unclear.

Empirical evidence supports the idea that plastic changes in behaviour can influence individual morphological or physiological traits (Benard, 2004; Dzialowski et al., 2003; Nielsen and Papaj, 2022; Relyea, 2001; Van Buskirk, 2002). Here, we tested the hypothesis that such trait interactions can shape direct carry-over effects. Specifically, we investigated whether plasticity in behavioural traits interferes with the direct carry-over of body size across metamorphosis in the sea rock-pool mosquito *Aedes mariae*. Mosquitoes undergo four distinct life stages across their life cycle, which include the egg, larval, pupal, and adult stages (Clements, 1992). The larval and pupal size – typically measured as body length or cephalo-thorax width (Bar and Andrew, 2013; Ukubuiwe et al., 2018) – are widely used as proxies for adult body size due to carry-over effects, which is in turn linked to ecologically relevant traits such as fecundity, longevity, dispersal ability, and mating success (Alto and Juliano, 2001; Armbruster and Hutchinson, 2002; Cator et al., 2010; Hatala et al., 2018; Maciel-De-Freitas et al., 2007; Villarreal et al., 2017). Hence, trait correlations across life stages can have direct implications for fitness and evolutionary trajectories.

In *Ae. mariae*, larval and pupal stages develop in the sea-rock pools of the supralittoral zone along the western Mediterranean coast (Coluzzi et al., 1974; Mastrantonio et al., 2015), which are extremely variable habitats, experiencing high salinity fluctuations from freshwater to hypersaline conditions over extremely short time frames (Rioux, 1958; Rosenfeld et al., 2019). Salinity is therefore an ecologically realistic and temporally dynamic stressor in this system, with known effects on larval physiology, growth and behaviour in several mosquito species (e.g. De Brito Arduino et al., 2015; Clark et al., 2004; Patrick and Bradley, 2000; Ramasamy et al., 2011; Silberbush et al., 2014). For example, in *Aedes aegypti*, increasing salinity reduces both pupal size and adult body mass (De Brito Arduino et al., 2015), whereas in the euryhaline *Ochlerotatus taeniorhynchus*, body mass increases with increasing salinity (Clark et al., 2004). Such contrasting responses highlight the role of species-specific plasticity and suggest that salinity can modulate developmental trajectories in diverse ways across mosquito taxa. Using an individual-based approach, we tracked larval and pupal development under two different salinity conditions (constant versus increasing water salinity). We measured larval activity and body size throughout metamorphosis to assess plasticity in these traits and performed path analysis to infer causal relationships between them during development to test the hypothesis that the interaction between behavioural and morphological plasticity can shape direct carry-over effects.

## RESULTS
### Plastic response of morphological and behavioural larval traits

Out of the initial 60 individuals, 53 were included in the analysis (C=27; S=26), while seven were excluded due to mortality before reaching the pupal stage, which prevented the collection of complete developmental measurements required for the analysis. The principal component analysis (PCA) on morphological measures showed that the first PCA component explained 76.9% of the total variance and discriminated against individuals on body size. The second component explained 13.9% of the variance and detected differences in body shape (Table S1).

The generalised linear model (GLM) conducted on larval body size (PC1) showed significant differences between treatments (Table 1). Fourth instar larvae grown in increasing salinity treatment were smaller than larvae in constant treatment (Fig. 1A). We found no effect of treatment on the larval shape (PC2; Table 1, Fig. 1B).

The activity behaviour of larvae varied significantly between treatments (Fig. 1C,D). Increasing salinity caused an increase in resting by 7.22% and reduced browsing behaviour (i.e. active feeding) by 6.31% compared to the constant treatment (Table 1).

### Path analysis
In constant salinity treatment, our path analysis revealed a strong positive impact of the larval size on the pupae cephalo-thorax (Fig. 2A,D). Resting behaviour did not influence pupae cephalo-thorax width (Fig. 2C,D) and did not depend on larval body size (Fig. 2B,D). Overall, bigger larvae resulted in bigger pupae.

Under increasing salinity conditions, the covariation patterns changed: cephalo-thorax width was no longer dependent on larval size (Fig. 2A,E). Moreover, in this salinity condition, the larval size negatively influenced the resting behaviour, meaning that smaller larvae performed more resting [and less browsing since there is a strong negative correlation between these two behaviours (r=−0.91, P<0.001; Fig. 2B,E)].

Finally, we refitted the structural equation model (SEM) analysis using the whole dataset, including treatment as a factor. Results identified one statistically significant change in the pathways between treatments: the strong positive relationship between larval body size and pupae cephalo-thorax width became non-significant with increasing salinity (Fig. 2D,E, Table 2). We replicated this test using a GLM with an interaction term (larval size×treatment) predicting pupal size. The interaction was highly significant (P<0.001), supporting the SEM result that the effect of larval size on pupal size was strongly modulated by salinity condition (Table S2).

## DISCUSSION
In this study, we measured larval activity and body size throughout metamorphosis to determine whether these traits exhibit plasticity under changing salinity conditions. We then performed a path analysis to infer causal relationships between body size and

**Table 1. Outcome of GLMs using treatment as a fixed factor**

| Trait | Coefficient | Standard error | *t*-value | *P*-value |
|---|---|---|---|---|
| Larval body size – PC1 | −2.583 | 0.532 | −4.849 | **<0.001** |
| Larval body shape – PC2 | −0.460 | 0.266 | −1.728 | 0.090 |
| Larval spontaneous activity–resting behaviour | 0.077 | 0.038 | 2.02 | **0.048** |
| Larval spontaneous activity– browsing behaviour | 9.876 | 3.914 | 2.523 | **0.014** |

Significant *P*-values are shown in bold.

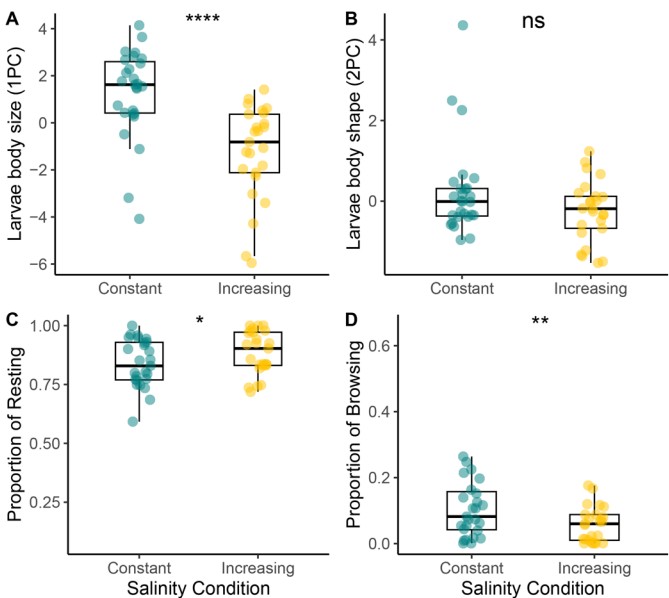

**Fig. 1. Larval phenotypic traits.** (A) PCA first component (larval size), (B) PCA second component (larval shape), (C) Proportion of time spent in resting and (D) browsing of fourth instar larvae under constant (C; *N*=27) and increasing salinity (Ş *N*=26) conditions. Points are individual observations. Significance levels: ****P*<0.001, ***P*<0.01, **P*<0.05, ns=*P*>0.05. Boxplots show median values (middle line), interquartile range (box), and range values, including some outliers (dots that extend beyond the min and max of the boxplot).

behaviour during development and to test the hypothesis that such trait interactions can shape direct carry-over effects. Overall, our results show that salinity influences plastic changes in both the morphology and behaviour of *Ae. mariae* larvae. Most importantly, we found that salinity affects trait covariation, which in turn influences body size development across metamorphosis, supporting our hypothesis.

First, we observed a plastic response to increasing salinity, resulting in smaller larval body size and reduced browsing behaviour (Fig. 1). These changes can be attributed to the energetic costs of osmoregulation. Several studies have shown that mosquito species inhabiting brackish or saline water maintain osmotic balance through different strategies, such as accumulating amino acids in the hemolymph (Garrett and Bradley, 1987; Patrick and Bradley, 2000), increasing the size of anal papillae (Surendran et al., 2018) or producing less permeable cuticles (Bradley, 1987). These mechanisms require substantial energy expenditure, which is traded off against resource allocation for body growth (Clark et al., 2004; De Brito Arduino et al., 2015; Nayar, 1969). Interestingly, our analysis of activity behaviour revealed that larvae in the increasing salinity treatment spent less time browsing (i.e. actively feeding) and more time resting compared to those in the constant salinity treatment. Since mosquito larvae filter and ingest water while resting at the surface, oral water intake in saltwater mosquito species may counteract osmotic water loss through the cuticle (Merritt et al., 1992). For instance, in *Culex tarsalis*, larvae transferred to higher salinity increased their drinking rate by over 50% compared to freshwater conditions (Patrick and Bradley, 2000). Accordingly, the combined effect of energy allocation to osmoregulation and behavioural plasticity in *Ae. mariae* likely contributes to its smaller body size under increasing salinity conditions.

Secondly, using path analysis to examine the correlation between larval traits in the two treatments, we found no interaction between

larval size and activity under constant salinity conditions. However, under increasing salinity, activity behaviour became size-dependent. Specifically, we observed a strong negative effect of larval body size on activity behaviour (Fig. 2B-E). This suggests that salinity conditions influence trait covariation. This finding aligns with existing literature, as it is well established that environmental conditions can shape not only the expression of individual traits but also their covariation, a phenomenon known as trait integration (i.e. the tendency of traits to vary together because they are developmentally or functionally linked) (Kasumovic, 2013; Nielsen and Papaj, 2022; Pigliucci, 2003; Schlichting, 1989). For instance, Relyea (2001) found that in multiple species of larval anurans, the patterns of trait covariation between morphological and behavioural traits varied depending on predator exposure; in predator-free environments, activity was positively correlated with body depth and width, a pattern that was partially maintained (between activity and body width) in the presence of *Umbra* predators but entirely absent in the presence of *Anax* predators. Similarly, a negative correlation between activity and tail depth was observed in the absence of predators and in the presence of *Umbra* individuals but disappeared in the presence of *Anax* predators.

As we move from the larval to the pupal stage, we found that the plasticity of phenotypic integration influences phenotypic development throughout metamorphosis. Specifically, under constant salinity conditions – where no correlation between larval size and activity was observed – we detected a strong, positive, and significant effect of larval size on pupal size. This indicates that larger larvae developed into larger pupae, demonstrating a direct carry-over effect. Conversely, under increasing salinity conditions, the emergence of size-dependent activity behaviour completely disrupted the relationship between larval size and pupal size, effectively breaking the direct carry-over effect. The observed disruption of the carry-over effect under increasing salinity conditions could also arise from direct responses to salinity during the pupal stage; however, this is unlikely, as we measured pupal traits within 2 h of pupation. Trait decoupling across metamorphosis can result from changes in genetic correlations between life stages or from differences in the environmental sensitivity of later-stage development (Aguirre et al., 2014; Fellous and Lazzaro, 2011; Gomez-mestre and Buchholz, 2006; Johansson et al., 2016; Mikolajewski et al., 2015). However, in the case of direct carry-over effects, constraints on decoupling are typically strong, as later-stage phenotypes are inherently shaped by developmental factors acting on the same traits earlier in life (Cheverud, 1984; Moore and Martin, 2019). Here, we provide evidence that interactions between plastic traits may serve as an alternative mechanism for decoupling in cases of direct carry-over. Our results may have broad relevance since many organisms with complex life cycles and metamorphic transitions, such as amphibians, aquatic insects, crustaceans, and echinoderms, develop under ecologically variable conditions. In these systems, larval environments can strongly influence later stages (Bouchard et al., 2016; De Block and Stoks, 2005; Evans et al., 2021; Garcia et al., 2017; Marshall and Connallon, 2023), but our findings suggest that environmental stress may also disrupt otherwise conserved developmental correlations within and between life stages.

## Conclusions

The extent to which one life stage is linked to others across metamorphosis remains a topic of debate. According to the 'adaptive decoupling hypothesis', the phenotype of one life stage

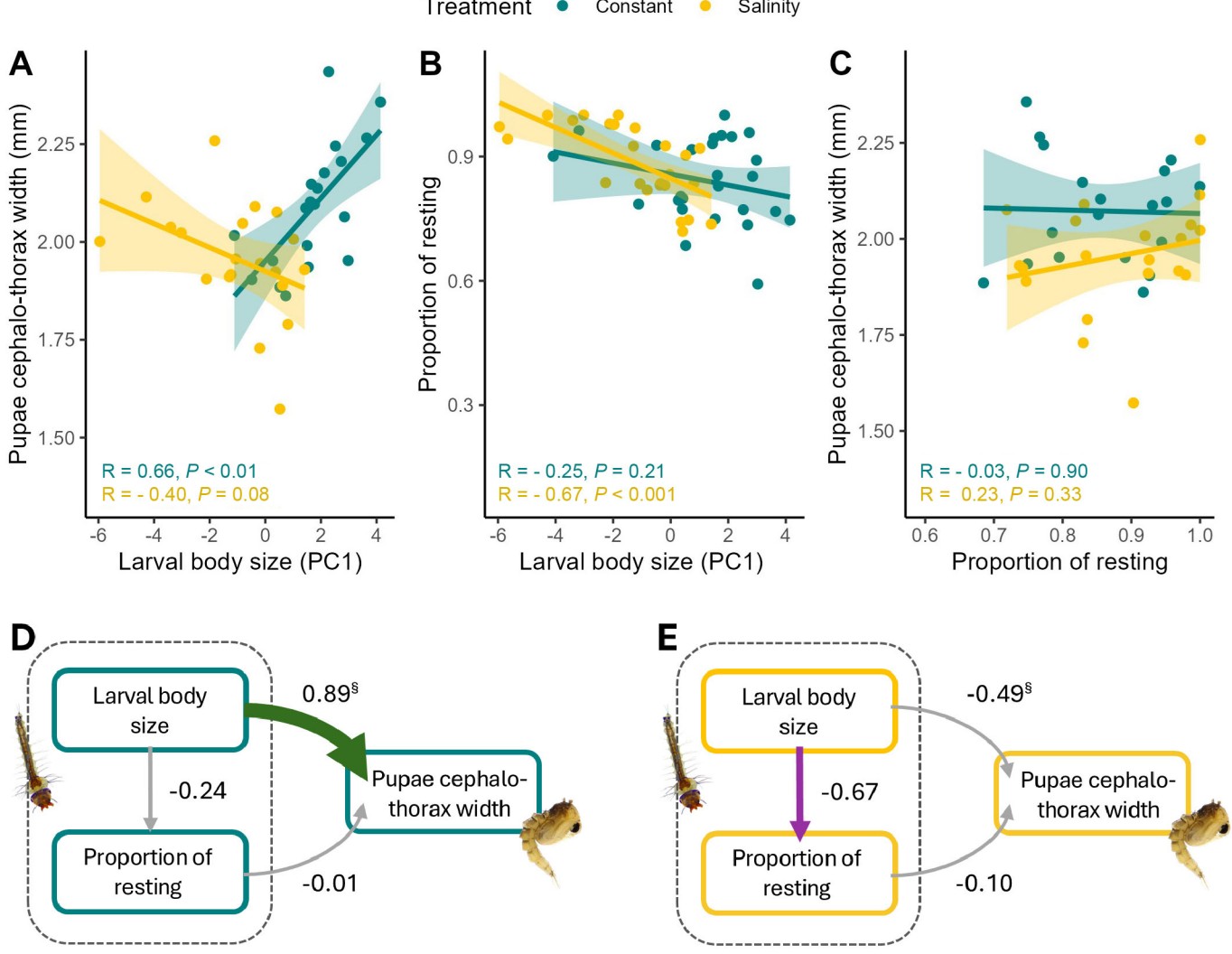

**Fig. 2. Trait correlations and structural equation models under constant and increasing salinity.** (A-C) Relationships between larval body size (PC1), pupal cephalo-thorax width, and resting proportion under two salinity conditions (constant and increasing salinity). Each plot shows the linear regression relationship for the two conditions, with constant salinity represented by blue and increasing by yellow points and lines. Shaded regions represent the 95% confidence intervals for each regression line. Pearson's correlation coefficient (R) and corresponding P-values are displayed for each condition within each panel. (D) Structural equation model (SEM) for constant treatment and (E) for increasing salinity treatment. Green and purple arrows represent significant (P<0.05) positive and negative paths, respectively, and grey arrows represent non-significant paths. Numbers next to the arrows are averaged effect sizes as standardised path coefficients; arrow widths reflect these standardised effect sizes. § represents paths with P≤0.05 in multigroup analysis (for exact P-values see Table 2). For the percentage of variance explained by response variables, non-standardised coefficient values and exact P-values of individual paths, see Table S3.

does not influence other stages, allowing traits to evolve independently across developmental transitions (Aguirre et al., 2014; Moran, 1994). Conversely, evidence of carry-over effects across a wide range of organisms with complex life cycles or those spanning multiple environments (Crean et al., 2011; De Block and Stoks, 2005; Norris and Taylor, 2005; Pechenik, 2006; Vonesh, 2005) suggests that metamorphosis does not represent an entirely

new beginning, but rather a process in which different life stages remain developmentally connected. Our findings reveal that the connection between life stages across metamorphosis is not fixed but can be plastically reshaped by environmental stressors, leading to a 'plastic' direct carry-over effect.

## MATERIALS AND METHODS
### Sampling and experimental conditions
Experiments were carried out on larvae of *Ae. mariae* obtained from eggs collected in July 2020 from supralittoral rock pools of San Felice Circeo, Italy (41°13′18.77″N, 13°4′5.51″E). We designed two experimental treatments based on field data collected during the reproductive season of the species. In the first treatment, individuals were maintained at a constant 50‰ salinity (50 g/l) throughout the experiment (constant condition, hereafter C). In the second treatment, larvae were exposed to increasing salinity from 50‰ to 150‰, with an increase of 10‰ per day throughout the experiment (salinity condition, hereafter S). The experiments were run in a climate chamber set to 26±1°C, 14 h light and 10 h dark regime.

**Table 2. Exact P-values of the multigroup analysis**

| Response | Predictor | P-value |
|---|---|---|
| Pupae cephalo-thorax width | Larval body size | **<0.001** |
| Pupae cephalo-thorax width | Proportion of resting | 0.785 |
| Proportion of resting | Larval body size | 0.173 |

This analysis implements a model-wide interaction in which every term in the model interacts with the grouping variable (i.e. constant versus increasing salinity). Significant P-value is shown in bold.

Experimental larvae were fed daily with 1 mg of cat food (Imam et al., 2014).

## Morphological and behavioural trait measurements

Second instar larvae were individually placed into plastic trays (12×12×7 cm) filled with 200 ml of tap water, previously salted with aquarium salt (Tetra Marine Seasalt). Tap water was left to stand for at least 24 h before use, allowing the chlorine to dissipate naturally. Then, we randomly assigned individuals to constant or salinity developmental treatments ($N$=30×treatment), and trays were maintained under strictly controlled and homogeneous temperature, humidity, and photoperiod conditions within the climate chamber. We placed larvae individually to track the development of each individual across its life cycle from larval stages to pupae.

We measured the fourth instar larvae's body size and spontaneous activity. Morphometric measures were obtained by taking digital pictures using a stereomicroscope Leica EZ4W at magnification 1× of all individuals within 2 h of the ecdysis. Subsequently, using the open-access software IMAGEJ we measured the width and length of the head, thorax, abdomen, and total body (Bar and Andrew, 2013). Larval spontaneous activity was analysed by video-recording larvae for 10 min in their housing containers using a camera Canon TG-6. Individuals were tested within 12 h from the ecdysis between 08:00 and 13:00. A single operator (G.C.), blind to the treatment, manually scored videos using the software BORIS (Behavioral Observation Research Interactive Software). Larval behaviours in the ethogram consisted of resting and browsing. During resting behaviour, the larva is positioned underwater or at the water surface with the respiratory siphon attached to the air-water interface and the body hanging obliquely into the water column. In browsing behaviour, the larva brushes the wall or the bottom of the container with its mouthparts and moves along the container surface (Merritt et al., 1992).

When larvae reached the pupal stage, we measured the pupal body size. Measurements were obtained from digital images, following the same procedure described for the larval stage. As for larvae, digital pictures were obtained within 2 h of pupation to avoid any substantial morphological changes that could occur in response to the pupal environment. This increased the likelihood that the measurements primarily reflected larval developmental conditions, rather than *de novo* plastic responses during the pupal stage.

We measured cephalo-thorax width (Bar and Andrew, 2013; Ukubuiwe et al., 2018) and determined the sex of individuals by checking the last cephalo-thorax segment (Clements, 1992). Sex cannot be reliably determined at the larval stage in mosquitoes. Therefore, we determined sex at the pupal stage and then used this information retrospectively in statistical analyses of larval traits.

## Data analysis

To find evidence of plastic response in larval morphology and spontaneous activity to salinity conditions, we carried out GLMs. For all models, we inspected residual plots to verify distributional assumptions. When assumptions of Gaussian models were not met, we fitted GLMs with alternative error distributions. The choice of distribution and link function was guided by the Akaike information criterion (AIC) (Table S4, Fig. S1). We applied PCA on fourth-instar larvae morphometric measures and subsequently performed GLM on the first two principal components separately as dependent variables. The analysis was run using the *glm* function in R with a Gaussian distribution and with identity link function for PCA components of larval morphology and larval resting behaviour. We applied GLMs with Gamma distribution and inverse link for browsing behaviour. We checked for the effect of sex because of the sexual dimorphism in this mosquito species, although it is usually not evident at the larval stage (Cordeschi et al., 2024); however, for models where there was no evidence for an effect of sex based on likelihood ratio test (LRT), we did not include this term in the final models (see Table S5).

To infer causal relationships between morphological and behavioural traits along the development, we performed path analysis using piecewise structural equation approach using *piecewiseSEM* package. SEM provide means to link multiple predictor and response variables in a single casual network, in which paths indicate hypothesised relationships between variables (Lefcheck, 2016). Structural equations modelling is especially useful when response variables can also act as predictor variables of another response variable – that is, when they mediate an indirect relationship. Since we hypothesised that the body size of early developmental stages might determine the body size of the successive stage and expected an influence of the activity behaviour in shaping body size throughout the development, we included three linear relationships in our path analysis (lm models). We fit two separate SEM for treatments since we hypothesised that the covariation patterns depend on water salinity. We used in the model only resting behaviour because of the significantly high correlation between resting and browsing measures (r=−0.91, $P$<0.001). The coefficients were standardised to allow comparison between variables.

Using the *multigroup* function, we performed a multigroup analysis to evaluate differences in path coefficients between salinity conditions models. This analysis implements a model-wide interaction in which every term in the model interacts with the grouping variable (i.e. constant versus increasing salinity). If the interaction is significant, the path is different between salinity conditions and is free to vary by group; if not, the path is constrained and takes on the estimate from the global dataset. To confirm whether the relationship between larval and pupal size varied across salinity treatments, we fitted an additional linear model that included an interaction term between larval size and salinity. All statistical analyses were performed in R version 4.4.2.

## Acknowledgements
The authors thank Nicole Giardiello and Alessandra Spanò for their technical help. The authors also thank the Editor and the two anonymous reviewers for their comments that improved the manuscript.

## Competing interests
The authors declare no competing or financial interests.

## Author contributions
Conceptualization: G.C., D.C., D.P.; Data curation: G.C., R.B., V.M.; Formal analysis: G.C.; Funding acquisition: D.C., D.P.; Investigation: G.C., R.B., V.M.; Methodology: G.C., D.P.; Project administration: D.P.; Resources: V.M., D.P.; Supervision: D.C., D.P.; Validation: D.C., D.P.; Writing – original draft: G.C.; Writing – review & editing: G.C., R.B., V.M., D.C., D.P.

## Funding
This research was funded by Ministero dell'Istruzione, dell'Università e della Ricerca (the Italian Ministry of University and Research; PRIN project 20222SYWHP). D.P. was supported by EU funding within the NextGenerationEU-MUR PNRR Extended Partnership initiative on Emerging Infectious Diseases (Project no. PE00000007, INF-ACT). Open Access funding provided by the Italian Ministry of University and Research (PRIN project:2017-HYBRIND MIUR - prot 017KLZ3MA) Deposited in PMC for immediate release.

## Data and resource availability
Data and the R code used for the statistical analysis of this work are available at: https://figshare.com/s/d8041600d4e6a4d698ab. All other relevant data and details of resources can be found within the article and its supplementary information.

## Peer review history
The peer review history is available online at https://journals.biologists.com/bio/lookup/doi/10.1242/bio.062033.reviewer-comments.pdf

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
