## [Peer Review File · Biology Open]

Phenotypic plasticity shapes carry-over effects in sea rock-pool mosquitoes

Giulia Cordeschi, Roberta Bisconti, Valentina Mastrantonio, Daniele Canestrelli, Daniele Porretta

DOI: 10.1242/bio.062033

Editor: Lewis Halsey

Review timeline

Original submission:	24 April 2025
Editorial decision:	3 July 2025
First revision received:	2 October 2025
Accepted:	3 October 2025

Original submission

First decision letter

MS ID#: bio.062033

MS Title: Phenotypic plasticity shapes carry-over effects in sea rock-pool mosquitoes

Authors: Giulia Cordeschi, Roberta Bisconti, Valentina Mastrantonio, Daniele Canestrelli, Daniele Porretta

I have now reached a decision on the above manuscript.

The reviewer reports are shown at the bottom of this email or can be accessed, together with a copy of this decision letter, by going to:

As you will see, the reviewers raised a number of substantial criticisms that prevent me from accepting the paper at this stage.

They suggest, however, that a revised version might prove acceptable, if you can address their concerns. If you think that you can deal satisfactorily with the criticisms on revision, I would be pleased to see a revised manuscript. We would then return it to the reviewers.

At this stage, we also ask you to ensure your manuscript complies with our formatting guidelines. Provided you are able to fully address the referees' comments, we are positive about publication of your paper (we accept over 95% of revision submissions) and therefore hope you won't mind any extra work involved in reformatting your manuscript at this point.

Please ensure that you clearly highlight all changes made in the revised manuscript. Please avoid using 'Tracked changes' in Word files as these are lost in PDF conversion.

I should be grateful if you would also provide a point-by-point response detailing how you have dealt with the points raised by the reviewers in the 'Response to Reviewers' box. Please attend to all of the reviewers' comments. If you do not agree with any of their criticisms or suggestions please explain clearly why this is so.

Reviewer 1

Comments for the author

Cordeschi et al. present a compelling study addressing fundamental concepts in evolutionary ecology using the sea rock pool mosquito. This organism serves as an interesting model system, well-suited for investigating developmental responses to potent environmental stressors such as drastically fluctuating salinity. The authors adeptly use this system to revisit prevailing assumptions that often treat carry-over effects as linear and fixed across environmental gradients. Instead, they propose that environmental stress can disrupt otherwise conserved developmental correlations, suggesting that plastic interactions among traits can decouple trait expression across life stages.

Their study possesses considerable conceptual depth and is highly relevant to modern ecology and evolutionary biology. To maximize its impact and ensure accessibility to a broader scientific audience, the manuscript would benefit significantly from a revision of its narrative style and clarity:

1. Certain phrasing, such as the title's "meets and moulds," while evocative, is less direct than typical scientific language and could be revised for clarity throughout the manuscript.
2. The Reviewer strongly encourages the authors to dedicate more space to thoroughly introducing, defining, and contextualizing fundamental yet complex concepts such as "phenotypic plasticity" and "carry-over effects." This would establish a clearer foundation for readers less familiar with this specific area of research.
3. Terms like "larval size-dependent behavioural plasticity," "trait interactions/integration," and "metamorphic boundary" are specialized and risk being cryptic to non-specialist readers. Providing clearer, more accessible explanations or definitions for such terms upon their introduction would be beneficial.

Other issues regarding study design and execution:

4. It's unclear how sex was determined or inferred for larvae. This ambiguity should be explicitly addressed in the methods, detailing the approach used and acknowledging any potential limitations or assumptions, as post hoc inference could introduce error.
5. Please specify whether larvae were randomly assigned across experimental units (e.g., trays) to control for potential subtle microenvironmental gradients. Explicit mention of randomization procedures or the use of a randomized block design would be valuable. If no randomization was implemented, please add a text for justification or limitation.
6. Potential non-normal distribution related: It is not clearly specified whether data transformations or GLMs with suitable error distributions were employed.
7. Provide comprehensive model diagnostics for all statistical models employed including checks for residual distribution, homoscedasticity, multicollinearity, and overdispersion (for GLMs). Detail any data transformations used and how model assumptions were met or issues addressed.
8. Expand the discussion section to more thoroughly address the potential generalizability of the findings, e.g. other organisms undergoing metamorphosis in ecologically variable environments.

Reviewer 2

Comments for the author

This manuscript explores the intriguing interplay between behavioural and morphological plasticity and its influence on carry-over effects across metamorphosis in *Aedes aegypti*. The authors present a well-motivated study that addresses how interactions among plastic traits can modulate or disrupt direct carry-over effects between developmental stages. The writing is clear and engaging, the study species is ecologically appropriate for the question, and the analyses are generally sound. I also commend the authors for taking a multivariate perspective on plasticity and contributing to our understanding of how trait integration is shaped by environmental stressors.

However, several key aspects of the manuscript could be strengthened to improve clarity and interpretation of the results. My main concerns relate to the need for greater clarity in hypothesis formulation, better alignment between background and experimental design, and stronger support for the core conclusions regarding statistical support for the purported carry-over effects. These are outlined below.

1. The manuscript repeatedly refers to "direct carry-over effects," but it is not always clear whether the discussion is meant to apply specifically to organisms with metamorphic life histories only, or more generally to all multistage development. The introduction could be improved by more explicitly defining the scope of the framework and clarifying whether the concepts and predictions apply exclusively to systems with metamorphosis (perhaps at first mention on line 42).

2. While the introduction does a good job outlining concepts of multivariate plasticity and trait integration, it provides little context for why salinity was chosen as the experimental stressor. Readers would benefit from a more targeted explanation of the ecological relevance of salinity variation in this system and from prior work demonstrating its physiological and behavioural effects on mosquitoes. This would help build stronger expectations for the observed trait changes.

3. The paper would be improved by providing more detail on how the larval and pupal traits measured here map onto the general framework outlined in the introduction. In particular, the pupal traits used to assess carry-over (e.g., cephalothorax width) are not clearly linked to fitness-relevant phenotypes or to examples of carry-over in other systems. Anchoring this more explicitly to the concepts laid out in the intro would help clarify the significance of the trait relationships under study.

4. A central claim of the paper is that salinity disrupts the relationship between larval and pupal size, decoupling developmental trajectories. While this is visually supported by SEM and correlation plots (Fig. 3), the statistical inference would be strengthened considerably by including explicit interaction terms in GLMs (e.g. larval size \times salinity). This would directly test whether the effect of larval size on pupal size is statistically different between treatments. Without such a test, the claim of disrupted carry-over remains somewhat under-supported.

5. One important limitation is that the larvae and pupae were maintained under the same salinity conditions, making it difficult to disentangle whether pupal size reflects larval conditions or pupal plasticity. To explicitly demonstrate that carry-over effects are disrupted by larval conditions, a cross-over design (e.g., transferring larvae to the opposite salinity at pupation) would be ideal. As it stands, it is not possible to rule out the possibility that the observed differences in pupal traits arise from direct responses to salinity during the pupal stage. This caveat should be acknowledged more explicitly in the discussion.

6. The methods state that larvae were reared in salted tap water, but it is unclear whether the water was dechlorinated. Given the sensitivity of mosquito larvae to water chemistry and the behavioural assays involved, please clarify whether any steps were taken to remove chlorine or condition the water prior to use.

Reviewer's Responses to Questions

Experimental quality

Does each figure have the proper controls?

If 'No', please indicate reasons in Comments for Author box below.

Reviewer #1:

- Yes

Reviewer #2:

- Yes

Were the data analyzed using appropriate statistical tests?

If 'No', please indicate reasons in Comments for Author box below.

Reviewer #1:

- Yes

Reviewer #2:

- No

Reproducibility

Were experiments performed using adequate number of biological replicates?

If 'No', please indicate reasons in Comments for Author box below.

Reviewer #1:

- Yes

Reviewer #2:

- Yes

Does the methods section provide sufficient detail to permit reproducibility?

If 'No', please indicate reasons in Comments for Author box below.

Reviewer #1:

- Yes

Reviewer #2:

- Yes

Completeness

Are the manuscript's conclusions supported by the data?

If 'No', please indicate reasons in Comments for Author box below.

Reviewer #1:

- Yes

Reviewer #2:

- No

Scholarship

Do the authors cite and discuss the merits of data that would argue for and against their conclusion?

If 'No', please indicate reasons in Comments for Author box below.

Reviewer #1:

- Yes

Reviewer #2:

- No

Does the manuscript title & abstract accurately reflect the contents of the manuscript, without hyperbole?

If 'No', please indicate reasons in Comments for Author box below.

Reviewer #1:

- Yes

Reviewer #2:

- Yes

First revision

Author response to reviewers' comments

Point-by-point response to reviewers

We thank both reviewers for their thoughtful and constructive comments, which have helped us to substantially improve the clarity, structure, and impact of our manuscript. We have carefully addressed all points raised, providing clarifications, adding new analyses where appropriate, and revising the text to enhance conceptual accessibility and methodological transparency. Below, we provide a detailed, point-by-point response to each comment.

Reviewer 1: *Cordeschi et al. present a compelling study addressing fundamental concepts in evolutionary ecology using the sea rock pool mosquito. This organism serves as an interesting model system, well-suited for investigating developmental responses to potent environmental stressors such as drastically fluctuating salinity. The authors adeptly use this system to revisit prevailing assumptions that often treat carry-over effects as linear and fixed across environmental gradients. Instead, they propose that environmental stress can disrupt otherwise conserved developmental correlations, suggesting that plastic interactions among traits can decouple trait expression across life stages.*

Their study possesses considerable conceptual depth and is highly relevant to modern ecology and evolutionary biology. To maximize its impact and ensure accessibility to a broader scientific audience, the manuscript would benefit significantly from a revision of its narrative style and clarity:

1. *Certain phrasing, such as the title's "meets and moulds," while evocative, is less direct than typical scientific language and could be revised for clarity throughout the manuscript.*

REPLY: We appreciate this suggestion. We have revised the title to a clearer and more direct form: "*Phenotypic plasticity shapes carry-over effects in sea rock-pool mosquitoes.*" In addition, we carefully revised the manuscript to simplify phrasing where needed, aiming for greater clarity and alignment with conventional scientific style.

2. *The Reviewer strongly encourages the authors to dedicate more space to thoroughly introducing, defining, and contextualizing fundamental yet complex concepts such as "phenotypic plasticity" and "carry-over effects." This would establish a clearer foundation for readers less familiar with this specific area of research.*

REPLY: We have now revised the Introduction to more thoroughly define and contextualize some complex concepts, including phenotypic plasticity (lines 37-44) and carry-over effects (lines 44-49).

3. *Terms like "larval size-dependent behavioural plasticity," "trait interactions/integration," and "metamorphic boundary" are specialized and risk being cryptic to non-specialist readers. Providing clearer, more accessible explanations or definitions for such terms upon their introduction would be beneficial.*

REPLY: We thank the reviewer for pointing this out. We have revised the manuscript to provide clearer explanations or short explanatory definitions when introducing specialized terms. (lines 24;79;198-199).

Other issues regarding study design and execution:

4. *It's unclear how sex was determined or inferred for larvae. This ambiguity should be explicitly addressed in the methods, detailing the approach used and acknowledging any potential limitations or assumptions, as post hoc inference could introduce error.*

REPLY: We thank the Reviewer for highlighting this point. Sex cannot be reliably determined at the larval stage in mosquitoes. We therefore determined sex at the pupal stage, where morphological characters are clearly diagnostic (Clements 1992), and then used this information retrospectively in the statistical analyses of larval traits. This is a standard approach in mosquito developmental studies and does not introduce error in the measurement of larval traits. We included this issue in the Methods (lines 285-287)

5. *Please specify whether larvae were randomly assigned across experimental units (e.g., trays) to control for potential subtle microenvironmental gradients. Explicit mention of randomization procedures or the use of a randomized block design would be valuable. If no randomization was implemented, please add a text for justification or limitation.*

REPLY: We thank the Reviewer for this comment. We have clarified in the Methods that each larva was kept individually in a tray and was randomly assigned to one of the two salinity treatments. All trays were maintained under strictly controlled and homogeneous conditions within the climate chamber, ensuring that potential microenvironmental gradients were negligible (lines 260-263).

6. *Potential non-normal distribution related: It is not clearly specified whether data transformations or GLMs with suitable error distributions were employed.*

REPLY: We have clarified in the Methods that we checked for model assumptions (normality and homoscedasticity) and applied GLMs with alternative error distributions when Gaussian assumptions were not met (e.g. Gamma distribution for browsing behaviour). The choice of error distribution and link function was guided by AIC and confirmed by visual inspection of residuals (lines 291-294).

7. *Provide comprehensive model diagnostics for all statistical models employed including checks for residual distribution, homoscedasticity, multicollinearity, and overdispersion (for GLMs). Detail any data transformations used and how model assumptions were met or issues addressed.*

REPLY: We thank the reviewer for this suggestion. We have now added a comprehensive section on model diagnostics in Supplementary materials.

8. *Expand the discussion section to more thoroughly address the potential generalizability of the findings, e.g. other organisms undergoing metamorphosis in ecologically variable environments.*

REPLY: We have expanded the Discussion to highlight the broader relevance of our findings in organisms with complex life cycles beyond mosquitoes (lines 224-230).

Reviewer 2: *This manuscript explores the intriguing interplay between behavioural and morphological plasticity and its influence on carry-over effects across metamorphosis in *Aedes aegypti*. The authors present a well-motivated study that addresses how interactions among plastic traits can modulate or disrupt direct carry-over effects between developmental stages. The writing is clear and engaging, the study species is ecologically appropriate for the question, and the analyses are generally sound. I also commend the authors for taking a multivariate perspective on plasticity and contributing to our understanding of how trait integration is shaped by environmental stressors. However, several key aspects of the manuscript could be strengthened to improve clarity and interpretation of the results. My main concerns relate to the need for greater clarity in hypothesis formulation, better alignment between background and experimental design, and stronger support for the core conclusions regarding statistical support for the purported carry-*

over effects. These are outlined below.

1. The manuscript repeatedly refers to "direct carry-over effects," but it is not always clear whether the discussion is meant to apply specifically to organisms with metamorphic life histories only, or more generally to all multistage development. The introduction could be improved by more explicitly defining the scope of the framework and clarifying whether the concepts and predictions apply exclusively to systems with metamorphosis (perhaps at first mention on line 42).

REPLY: We thank the reviewer for this insightful comment. We have revised the Introduction to clarify that carry-over effects are a general phenomenon that can occur in a wide range of biological contexts, including both metamorphic and non-metamorphic systems. We have also specified that our focus is on direct carry-over effects in organisms with complex life cycles involving metamorphosis, where such effects are particularly intriguing due to the presumed developmental decoupling between stages (lines 44-54).

2. While the introduction does a good job outlining concepts of multivariate plasticity and trait integration, it provides little context for why salinity was chosen as the experimental stressor. Readers would benefit from a more targeted explanation of the ecological relevance of salinity variation in this system and from prior work demonstrating its physiological and behavioural effects on mosquitoes. This would help build stronger expectations for the observed trait changes.

REPLY: In the revised Introduction, we have expanded the ecological rationale for using salinity as an experimental treatment (lines 97-105).

3. The paper would be improved by providing more detail on how the larval and pupal traits measured here map onto the general framework outlined in the introduction. In particular, the pupal traits used to assess carry-over (e.g., cephalothorax width) are not clearly linked to fitness-relevant phenotypes or to examples of carry-over in other systems. Anchoring this more explicitly to the concepts laid out in the intro would help clarify the significance of the trait relationships under study.

REPLY: Thank you for the suggestion. We now explicitly explain the relevance of the larval and pupal traits we measured (lines 86-93).

4. A central claim of the paper is that salinity disrupts the relationship between larval and pupal size, decoupling developmental trajectories. While this is visually supported by SEM and correlation plots (Fig. 3), the statistical inference would be strengthened considerably by including explicit interaction terms in GLMs (e.g. larval size \times salinity). This would directly test whether the effect of larval size on pupal size is statistically different between treatments. Without such a test, the claim of disrupted carry-over remains somewhat under-supported.

REPLY: We thank the reviewer for this helpful comment. We would like to clarify that the multigroup SEM inherently tests for interactions between path coefficients and the grouping variable (treatment). The *multigroup* function in the *piecewiseSEM* package fits a global model with interaction terms for each path, and tests whether path coefficients differ significantly between groups. Nevertheless, to enhance clarity and provide further support for the results, we have now explicitly tested the interaction between larval size and condition in a separate GLM with pupal size as the dependent variable (lines 149-152 and 322-324). This additional analysis confirms a significant interaction between larval size and condition (see Table S2 in the Supplementary Materials), strengthening the interpretation of our SEM results.

5. One important limitation is that the larvae and pupae were maintained under the same salinity conditions, making it difficult to disentangle whether pupal size reflects larval conditions or pupal plasticity. To explicitly demonstrate that carry-over effects are disrupted by larval conditions, a cross-over design (e.g., transferring larvae to the opposite salinity at pupation) would be ideal. As it stands, it is not possible to rule out the possibility that the

observed differences in pupal traits arise from direct responses to salinity during the pupal stage. This caveat should be acknowledged more explicitly in the discussion.

REPLY: We appreciate the reviewer's point and agree that, in general, carry-over effects can be confounded when conditions remain constant across life stages. However, in our study, pupal size was measured within two hours of metamorphosis, to avoid any significant morphological changes due to the pupal environment could occur. Therefore, pupal size in our experiment reflects mainly conditions experienced during the larval stage, supporting our interpretation of carry-over effects driven by larval traits rather than pupal plasticity. We have added a clarification to the Methods section to make this explicit (lines 279-283) and we acknowledged it in the discussion (lines 215-217).

6. The methods state that larvae were reared in salted tap water, but it is unclear whether the water was dechlorinated. Given the sensitivity of mosquito larvae to water chemistry and the behavioural assays involved, please clarify whether any steps were taken to remove chlorine or condition the water prior to use.

REPLY: We confirm that all tap water used in the experiment was left to rest for at least 24 hours before use to allow chlorine to dissipate naturally. This step was taken to avoid potential effects of residual chlorine on larval development and behaviour. We have now clarified this point in the Methods section (lines 259-260).

Second decision letter

MS ID#: bio.062033R1

MS Title: Phenotypic plasticity shapes carry-over effects in sea rock-pool mosquitoes

Authors: Giulia Cordeschi, Roberta Bisconti, Valentina Mastrantonio, Daniele Canestrelli, Daniele Porretta

I have read through and considered your extensive rebuttal document, and in turn I am happy to tell you that your manuscript has been accepted for publication in Biology Open, pending our standard publication integrity checks. It was accepted on 3rd October 2025.